# The Clinical Dilemma of Esophagogastroduodenoscopy for Gastrointestinal Bleeding in Cardiovascular Disease Patients: A Nationwide-Based Retrospective Study

**DOI:** 10.3390/jcm11133765

**Published:** 2022-06-29

**Authors:** Chao-Feng Chang, Wu-Chien Chien, Chi-Hsiang Chung, Hsuan-Hwai Lin, Tien-Yu Huang, Peng-Jen Chen, Wei-Kuo Chang, Hsin-Hung Huang

**Affiliations:** 1Division of Gastroenterology and Hepatology, Department of Internal Medicine, Tri-Service General Hospital, National Defense Medical Center, Taipei City 114, Taiwan; taiwanvincent777@gmail.com (C.-F.C.); redstone120@gmail.com (H.-H.L.); teinyu.chun@msa.hinet.net (T.-Y.H.); pjc.taiwan@gmail.com (P.-J.C.); weikuohouse@hotmail.com (W.-K.C.); 2Department of Medical Research, Tri-Service General Hospital, National Defense Medical Center, Taipei City 114, Taiwan; chienwu@mail.ndmctsgh.edu.tw; 3School of Public Health, Tri-Service General Hospital, National Defense Medical Center, Taipei City 114, Taiwan; g694810042@gmail.com; 4Taiwanese Injury Prevention and Safety Promotion Association, Tri-Service General Hospital, National Defense Medical Center, Taipei City 114, Taiwan; 5Division of Gastroenterology, Department of Internal Medicine, Cheng Hsin General Hospital, National Defense Medical Center, Taipei City 114, Taiwan

**Keywords:** esophagogastroduodenoscopy, gastrointestinal bleeding, cardiovascular disease

## Abstract

Performing esophagogastroduodenoscopy (EGD) in recently occurring peri-coronary artery disease (CAD) accident settings is always a dilemma. This study used the Taiwan National Health Insurance Research Database to identify patients with CAD and gastrointestinal bleeding who had received EGD or not between 2000 and 2013.The final population included in this study was 15,147 individuals, with 3801 individuals having received EGD (study cohort group) and 11,346 individuals not having received EGD (comparison cohort group). We initially performed a sensitivity test for CAD recurrence-related factors using multivariable Cox regression during the tracking period. A relatively earlier EGD intervention within one week demonstrated a lower risk of CAD recurrence (adjusted HR = 0.712). Although there were no significant differences in the overall tracking period, the adjusted HR of CAD recurrence was still lower in patients in the EGD group. Furthermore, our findings revealed that there were no remarkably short intervals to CAD recurrence in the study group. The Kaplan–Meier survival curve demonstrated that individuals who underwent EGD were not associated with a significantly increased CAD recurrence rate compared with the control (Log-rank test, *p* = 0.255). CAD recurrence is always an issue in recent episodes of peri-CAD accident settings while receiving EGD. However, there is not a higher risk in comparison with the normal population in our study, and waiting periods may not be required.

## 1. Introduction

Patients suffering from coronary artery disease (CAD) have significantly high mortality rates within a period of one month. Antithrombotic agents are a common method of treating CAD as they can decrease the incidence of subsequent CAD events; however, these drugs are more likely to increase bleeding tendency, especially upper gastrointestinal bleeding (UGIB). Approximately 1–4% of patients have concurrent CAD and UGIB, and up to 7% of patients develop sustained nosocomial GI bleeding following a PCI condition [1].

It is always important to consider the indications and contraindications when performing an endoscopic procedure so as to decide the exact timing of the endoscopy. A common dilemma is when patients have unstable hemodynamic status and a serious concurrent comorbidity, especially when it comes to cardiopulmonary problems. In clinical practice, perivascular accident settings with EGD are always more alarming to clinicians. This is because there are no guidelines to clearly define indications and contraindications in these cases. Cardiopulmonary side effects account for more than 50% of all complications and causes of death. Among endoscopic procedures, esophagogastroduodenoscopy (EGD) induces significant stress on the cardiopulmonary system, and the associated complications include hypertension and hypoxia. Fluctuations in blood pressure in approximately 40% of patients and unstable oxygenation with an oxygen saturation reduction in up to 70% of patients have been noted. Additionally, stress related to arrhythmias may result in cardiac ischemia in which the ECG shows ST-segment depression [2,3,4]. Furthermore, with the above urgent condition and possible unstable vital signs, analgesic agents for conscious sedation to relieve patient stress are far from appropriate.

Several studies in the current literature have investigated the utility of EGD in patients with concurrent vascular accident problems. Manifestations of severe complications while performing EGD are relatively infrequent [5]. However, peri-procedural complications are a great concern. Therefore, the purpose of our study was to use a national, large-population data sample to analyze CAD recurrence, mortality rates, and associated parameters in patients with CAD who received or did not receive EGD.

## 2. Methods

### 2.1. Data

The Taiwan National Health Insurance Research Database (NHIRD) was established in 1995, and the Taiwan National Health Insurance Administration Ministry of Health and Welfare (Taipei, Taiwan, China) provides a number of medical services, including inpatient, outpatient, and emergency services to >99% of the population in Taiwan. In the present study, data from the NHIRD were used. The investigation protocols were approved by the official peer review committee of the Tri-Service General Hospital (Taipei, Taiwan, China). The diagnoses were made according to the International Classification of Diseases, 9th Revision, Clinical Modification (ICD-9-CM) [6].

### 2.2. Study Cohort

A retrospective cohort design was used, and we selected outpatient and inpatient data between 1 January 2000 and 31 December 2013 from the NHIRD in Taiwan. The selected patient cases had concurrent health problems such as cardiovascular accident (CAD) and (UGIB). The included case group defined CAD by ICD-9-CM 410–414 and UGIB by ICD-9-CM 53X.0, 53X.2, 53X.4, 53X.6 (X = 1–4), 535.X1. We excluded patients for whom CAD/UGIB diagnosis was performed before the index date and patients under 20 years old. We also excluded cases where diagnosis of CAD/UGIB was performed before 1 January 2000. Furthermore, we excluded cases with unknown gender.

Initially, data from 16,482 patients were collected, of which 1335 individuals were excluded. Finally, our study included data from 15,147 patients. A total of 3801 individuals had received EGD as opposed to 11,346 individuals who had not. Next, we used a 4-fold propensity-score matching by gender, age, and index date. The 218 individuals who had received EGD within one month were defined as the study cohort group. In contrast, the 872 individuals without EGD were defined as the comparison cohort group (Figure 1). Notable variables included age, sex, and comorbidities of diabetes mellitus (DM) type II (ICD-9-CM 250), hypertension (ICD-9-CM 401-405), chronic obstructive pulmonary disease (COPD) (ICD-9-CM 491-493, 406), dyslipidemia (ICD-9-CM 272.0, 272.2, 272.4), and cancer (ICD-9-CM-140-208). The outcomes between these two groups were compared by balancing the above characteristics, follow-up durations, and survival status at the end of the tracking period, 31 December 2013. Thirteen individuals in the study cohort group and sixty-four individuals in the comparison cohort group experienced CAD recurrence within one month. When evaluating follow-up duration and survival status, we further analyzed other parameters, including urbanization level and level of care. In addition to CAD recurrence as the primary end point, we analyzed the severity by other factors: whether patients were admitted to an intensive care unit or were under mechanical ventilation or for the use of vasoconstrictors.

### 2.3. Statistical Analysis

We conducted all analyses by SPSS software (version 18; SPSS, Inc., Chicago, IL, USA). The χ^2^ and Fisher’s exact test were used for analysis of categorical variables, such as sex and comorbidities. The Student’s *t*-test was used for continuous variables, such as age, and the data are presented as mean ± standard error of the mean. Multivariate Cox regression was used to adjust the independent variables and to determine the association between each variable and CAD recurrence in one month. Additionally, CAD recurrence with different tracking periods and other covariates of outcomes were further analyzed by multivariable Cox regression stratified by EGD. Hazard ratios (HR) and 95% confidence intervals (CI) were used to evaluate the relative risks between each variable. Mean ± standard error of days to CAD recurrence were further investigated between the two groups. The Kaplan–Meier test was conducted to identify the cumulative survival of CAD with UGI bleeding to determine the statistical significance between groups.

## 3. Results

The clinical characteristics of the patients included in the present study are shown in Table 1. A total of 218 (20.00%) underwent EGD in the study cohort group, and 872 (80.00%) patients without EGD comprised the comparison cohort group. Following adjustment of variables, there were no statistical differences in the clinical characteristics between the study cohort group and the comparison cohort group. The distribution of gender, age, insurance premium, DM type II, HTN, COPD, dyslipidemia, and cancer between the two groups (with and without EGD) were similar. The mean age was 70.28 ± 12.08 years in patients with EGD and 71.12 ± 10.81 in patients without EGD. Males outnumbered females in both groups at the end of follow-up (58.26%). We identified that the insurance premiums were mostly less than New Taiwan dollar (NTD) 18,000, but this finding did not reach statistical significance (*p* = 0.328). There was no statistical difference in therapeutic variables hemostasis and endoscopic varices ligation (*p* = 0.560 and 0.844, respectively). Antiplatelet agent use between the two groups (with and without EGD) showed no significance (*p* = 0.172 and 0.221). Similarly, the urbanization level (from the highest to the lowest) was also not significant (*p* = 0.012). In contrast, the characteristic level of care was statistically different between the two groups (with more patients without receiving EGD). A greater number of patients with CAD received EGD in regional and local hospitals (Table 1). Additionally, our findings revealed that initial characteristics, including CAD recurrence and mortality rates within one month (*p* = 0.556 and 0.715, respectively) and length of days (*p* = 0.664), were not significantly different between the two groups. The average length of hospitalization was 70.30 ± 98.71 days (70.95 ± 95.10 and 67.70 ± 112.21 with and without EGD, respectively). Other factors included whether patients were admitted to an intensive care unit, underwent mechanical ventilation, or were administered vasoconstrictors, and these factors were further analyzed in our study and no significant difference was found in initial characteristics (*p* = 0.867, 0.867, and 0.727, respectively).

Multivariable Cox regression analysis on CAD recurrence within one month revealed no statistical significance in all variables examined, namely EGD, gender, age group, DM type II, HTN, COPD, CKD, dyslipidemia, cancer, urbanization level, and level of care. Patients in the EGD group were relatively less likely to experience recurrent CAD (adjusted HR = 0.855, *p* = 0.411). Furthermore, male patients had a relatively higher risk of CAD recurrence than female patients (adjusted HR = 1.052, *p* = 0.382). The elderly group of patients had an average higher risk of CAD recurrence; however, there were no significant differences among different age groups. With respect to variables of comorbidity (DM type II, HTN, COPD, dyslipidemia, and cancer), HTN and COPD were more likely to induce CAD recurrence (adjusted HR = 1.172 and 1.653, *p* = 0.424 and 0.598, respectively) as was the relatively higher urbanization level (adjusted HR = 1.503); however, there were no significant differences among different levels of urbanization. Furthermore, hospital centers and regional hospitals were more likely to be associated with CAD recurrence (adjusted HR = 2.986 and 1.872, respectively), but there were no significant differences among different levels of care (Table 2).

A sensitivity test was then performed pertaining to the factors of CAD recurrence by using multivariable Cox regression during the tracking period. Compared to patients without EGD, a lower adjusted HR was found in patients in the EGD group. EGD intervention within one month was the reference time point, and we observed that relatively earlier EGD intervention within one week had a lower risk of CAD recurrence (adjusted HR = 0.712). Moreover, we observed that when the intervention occurred at a later time, the adjusted HR became relatively higher. For instance, the adjusted HR of EGD interventions at 2 weeks, 3 weeks, and 1 month were 0.775, 0.834, and 0.855, respectively. Although there were no significant differences in the overall tracking period, the adjusted HR of CAD recurrence was still lower in patients in the EGD group (Table 3). In addition to CAD recurrence, other covariates that may associate with outcome were further analyzed by multivariable Cox regression, and increased adjusted HRs for ICU, mechanical ventilation, and vasoconstrictor use were found in EGD-receiving groups (adjusted HR = 1.298, 1.134, and 1.560, respectively). However, there was no statistical significance (*p* = 0.303, 0.486, and 0.762, respectively) (Table 4).

The days to CAD recurrence in one month between the two groups with and without EGD were 19.00 and 14.00, respectively, and the total average days to CAD recurrence in one month were 15.44. There were no remarkably short intervals to CAD recurrence in the study group (Table 5).

The Kaplan–Meier survival curve was used to analyze the cumulative survival of CAD recurrence. It was demonstrated that patients who underwent EGD were not associated with a significantly increased CAD recurrence rate compared with the control (Log-rank test, *p* = 0.255) (Figure 2).

## 4. Discussion

Most CAD cases showed improvement in the 28-day mortality and incidence rates of recurrent CAD episodes after increased use of fibrinolytic agents. Use of antiplatelet agents and heparin increased the risk of bleeding, especially UGI bleeding, which is usually caused by antiplatelet agents. However, discontinuation of these agents in UGIB is a significant concern for the development of CAD recurrence; hence, these drugs should be re-prescribed as soon as possible [7]. Clinically, UGIB can be divided into overt and occult bleeding. Patients with overt UGI bleeding commonly present with hematemesis and melena and are prone to developing signs of active bleeding. Hence, EGD results in remarkably positive effects and outcomes. In contrast, patients with occult bleeding may not be confronted with high-risk conditions. Cases with occult bleeding are at a relatively lower risk of requiring urgent endoscopy and discontinuation of anticoagulant therapy. As EGD is believed to be liable to lead to CAD development, gastroenterologists may be reluctant to perform EGD in the urgent situation of UGIB [8,9]. Whether CAD patients, who are prone to experiencing cardiopulmonary complications, should receive EGD or not depends on their clinical status. Among patients with epicardial coronary disease undergoing UGD, up to 16% are bound to show electrocardiographic evidence of periprocedural CAD events [10]. Endoscopic evaluation carries a higher-than-average risk in patients with recent CAD [11]. The severe endoscopic complications rate when performing EGD in acute myocardial infraction is approximately 1%, which is 10 times higher than routine EGD [1]. It was postulated that approximately 42% of patients may suffer from silent ischemia, which has been correlated to heartrate during EGD; thus, administration of β-blockers may be beneficial in this condition [9].

To our knowledge, this is the first large-scale study to explore the EGD procedure in recently-diagnosed CAD patients. We observed that EGD is a safe and beneficial procedure in relatively stable patients without unnecessary delays. The latest randomized controlled trial of early endoscopy for UGIB in CAD patients postulated that there was not a higher complication rate for EGD as compared with medication alone [12]. However, being different from other common severe complications such as gastrointestinal perforation or hemorrhage, cardiopulmonary conditions are a major concern, especially in CAD patients with decreased cardiopulmonary tolerance. The complication rate of EGD when performed on day 0 was higher than that performed after 24 h in the hospital setting, but endoscopy is more likely to be required sooner in sicker patients. Early endoscopy (more than 6 h and less than 24 h) provides superior timing compared to emergent EGD (less than 6 h), especially in nonvariceal bleeding conditions [8,13]. Rather than urgent UGD, a waiting period for later UGD is rarely mentioned and depends on patient condition. However, some studies suggest that it may be reasonable to wait up to a week after MI before performing EGD. Moreover, clinicians may have adequate time to perform fluid resuscitation, blood transfusion, and provide effective medication [14]. In our study, we performed a sensitivity test for different timings (weekly intervals) to analyze CAD recurrence using multivariable Cox regression. Our findings revealed that there was no significant difference between different timings using “weeks” as the unit for the waiting period. However, we did observe that the adjusted HR when performing UGD within one month was 0.855, which is slightly higher compared to 1, 2, and 3 weeks (Table 3). CAD recurrence events were not correlated with a waiting period for observation in clinical practice. Furthermore, we analyzed days to CAD recurrence: the number of days to CAD recurrence for the total population was 15.47 ± 9.45 days; with EGD and without EGD the figures were 19.50 ± 2.12 days and 14.93 ± 9.96 days, respectively (Table 5). In the CAD study group, CAD events occurred the most within 2 and 3 weeks, and there was no significant difference between the two groups. Unstable hemodynamic conditions a few weeks after CAD recurrence may deteriorate the cardiopulmonary condition of the patient. An observation period of 14–21 days after CAD recurrence is still important when performing EGD in CAD patients. Hence, a comprehensive evaluation before EGD is more important than performing endoscopy immediately. In our observation, there were increased risks with other covariates such as intensive care unit admission, mechanical ventilation, or the use of vasoconstrictors, but no statistical significance was observed. Although EGD was an important tool in UGIB diagnosis, the sicker patients with EGD intervention led to more complications, especially with multiple comorbidities. Risk stratification and gastrointestinal pathology confirmation were essential for selection of the patients to receive EGD [15].

International consensus recommendations on the management of patients with nonvariceal upper GI bleeding indicate that EGD should not be delayed for more than 24 h except in certain high-risk patients, such as those with acute coronary syndrome or a perforation [7]. With the recent advances in endoscopic techniques, the overall complication rates of EGD in clinical practice have been reduced from 0.13% to 0.08% [16,17,18,19,20]. However, performing endoscopy in peri-CAD accident settings, such as those related to insurance premium, urbanization level, and level of care, remains worrisome. In our study case, insurance premiums mostly ranged between NTD 18,000–34,999. However, this parameter was not found to be significant when performing endoscopy in the peri-CAD accident settings. Urgent endoscopic intervention was not correlated with increasing the cost of patient’s care among CAD patients. The ratio of CAD patients receiving EGD in hospital centers was relatively lower than those for regional and local hospitals. It was thought that CAD patients in hospital centers may suffer from multiple morbidities, and criteria for performing EGD would be stricter in consideration of the respective risk.

Despite efforts to control confounding factors, there is a number of limitations in the present study. Firstly, the information obtained from the NHIRD regarding patient characteristics was lacking in terms of detailed severity of CAD, medications used, and treatment modalities. Secondly, thorough information regarding the diagnosis of patients with CAD with UGIB was not disclosed in detail (e.g., hemoglobulin, coagulability). Thirdly, despite review from a specialist, there was potential bias due complicated comorbidities being missed. Further prospective studies may be required due to the retrospective nature of this observational study.

## 5. Conclusions

Comprehensive studies before the investigation of EGD in concurrent UGIB patients remain important. Different from common complications, cardiopulmonary complications should be more alarming to endoscopists. When EGD is performed, CAD recurrence is always an issue in recent episodes of peri-CAD accident settings while receiving care management. However, our retrospective study reveals that there this condition is not associated with an increased risk compared to normal populations, and a waiting period may not be required. EGD investigations should be based on an individual basis.

## Figures and Tables

**Figure 1 jcm-11-03765-f001:**
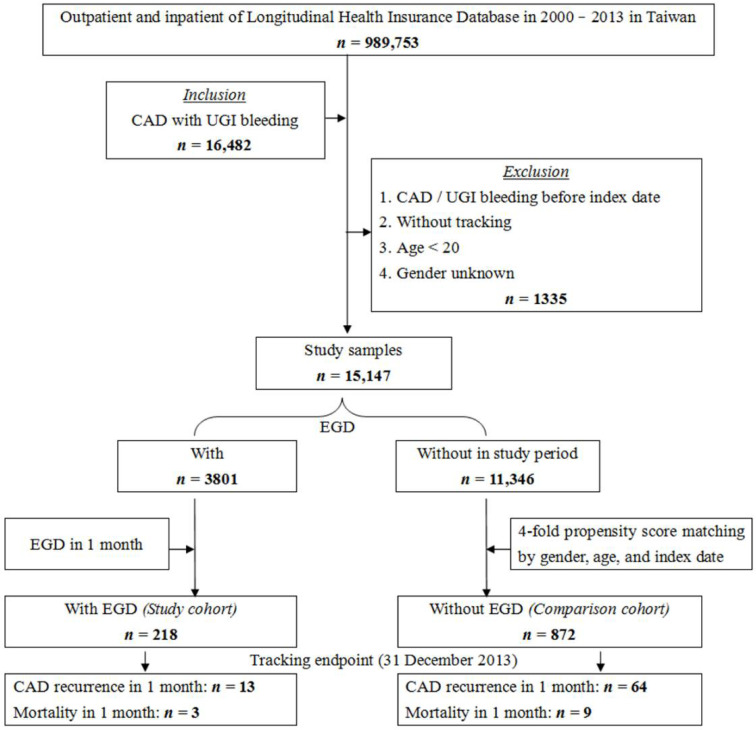
Flowchart of the study. The abbreviations: coronary artery disease (CAD); esophagogastroduodenoscopy (EGD); upper gastrointestinal (UGI).

**Figure 2 jcm-11-03765-f002:**
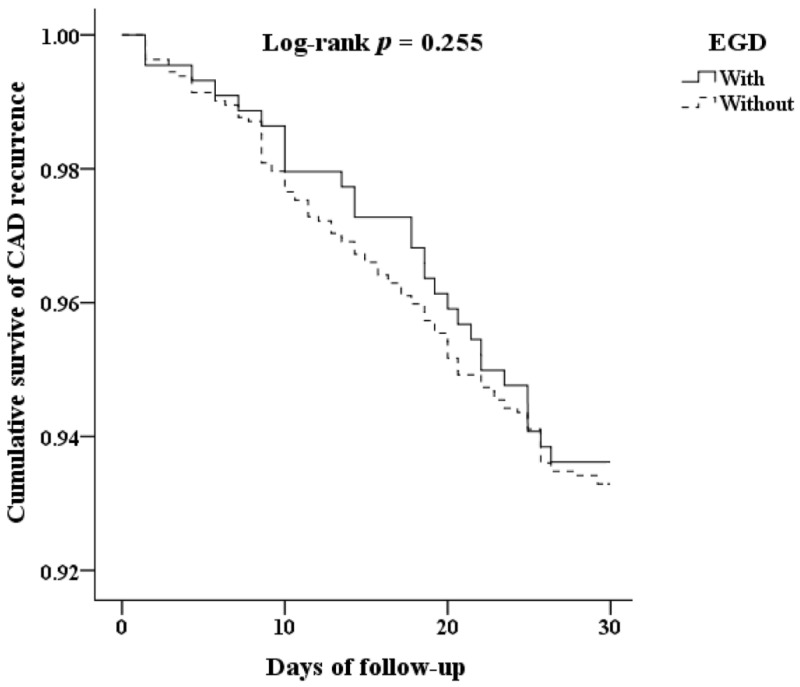
Kaplan–Meier for cumulative survival among CAD with UGI bleeding, aged 20 and over stratified by EGD with log-rank test. The abbreviations: coronary artery disease (CAD); upper gastrointestinal (UGI); esophagogastroduodenoscopy (EGD).

**Table 1 jcm-11-03765-t001:** Characteristics of study.

EGD	Total		With		Without		*p*
Variables	*n*	%	*n*	%	*n*	%	
Total	1090		218	20.00	872	80.00	
Gender							0.999
Male	635	58.26	127	58.26	508	58.26	
Female	455	41.74	91	41.74	364	41.74	
Age (yrs)	70.45 ± 11.84		70.28 ± 12.08		71.12 ± 10.81		0.349
Insured premium (NTD)							0.328
<18,000	1075	98.62	217	99.54	858	98.39	
18,000–34,999	15	1.38	1	0.46	14	1.61	
≥35,000	0	0.00	0	0.00	0	0.00	
DM type II							0.055
Without	719	65.96	156	71.56	563	64.56	
With	371	34.04	62	28.44	309	35.44	
HTN							0.024
Without	667	61.19	148	67.89	519	59.52	
With	423	38.81	70	32.11	353	40.48	
COPD							0.341
Without	1045	95.87	212	97.25	833	95.53	
With	45	4.13	6	2.75	39	4.47	
Dyslipidemia							0.081
Without	1047	96.06	214	98.17	833	95.53	
With	43	3.94	4	1.83	39	4.47	
Cancer							0.010
Without	985	90.37	207	94.95	778	89.22	
With	105	9.63	11	5.05	94	10.78	
Hemostasis							0.560
Without	919	84.31	181	83.03	738	84.63	
With	171	15.69	37	16.97	134	15.37	
EVL							0.844
Without	1028	94.31	205	94.04	823	94.38	
With	62	5.69	13	5.96	49	5.62	
Clopidogrel							0.172
Without	857	78.62	164	75.23	693	79.47	
With	233	21.38	54	24.77	179	20.53	
Aspirin							0.221
Without	689	63.21	130	59.63	559	64.11	
With	401	36.79	88	40.37	313	35.89	
Urbanization level							0.012
1 (Highest)	347	31.83	74	33.94	273	31.31	
2	507	46.51	86	39.45	421	48.28	
3	62	5.69	21	9.63	41	4.70	
4 (Lowest)	174	15.96	37	16.97	137	15.71	
Level of care							<0.001
Hospital center	402	36.88	56	25.69	346	39.68	
Regional hospital	486	44.59	99	45.41	387	44.38	
Local hospital	202	18.53	63	28.90	139	15.94	
CAD recurrence in 1 month							0.556
Without	1013	92.94	205	94.04	808	92.66	
With	77	7.06	13	5.96	64	7.34	
Mortality in 1 month							0.715
Without	1078	98.90	215	98.62	863	98.97	
With	12	1.10	3	1.38	9	1.03	
Length of days	70.30 ± 98.71	70.95 ± 95.10	67.70 ± 112.21	0.664			
ICU in 1 month							0.867
Without	1081	99.17	216	99.08	865	99.20	
With	9	0.83	2	0.92	7	0.80	
Mechanical ventilation in 1 month							0.867
Without	1081	99.17	216	99.08	865	99.20	
With	9	0.83	2	0.92	7	0.80	
Vasoconstrictors in 1 month							0.727
Without	1036	95.05	206	94.50	830	95.18	
With	54	4.95	12	5.50	42	4.82	

*p:* Chi-square/Fisher exact test on category variables and t-test on continue variables. The abbreviations: esophagogastroduodenoscopy (EGD); New Taiwan dollar (NTD); diabetes mellitus (DM); hypertension (HTN); chronic obstructive pulmonary disease (COPD); endoscopic variceal ligation (EVL); coronary artery disease (CAD); intensive care unit (ICU).

**Table 2 jcm-11-03765-t002:** Factors of CAD recurrence in one month by using multivariable Cox regression.

Variables	Adjusted HR	95% CI	95% CI	*p*
EGD				
Without	Reference			
With	0.855	0.793	1.352	0.411
Gender				
Male	1.052	0.599	4.578	0.382
Female	Reference			
Age group (yrs)	1.372	0.956	1.981	0.392
Insured premium (NTD)				
<18,000	Reference			
18,000–34,999	0.000	-	-	0.999
≥35,000	-	-	-	-
DM type II				
Without	Reference			
With	0.952	0.446	2.240	0.789
HTN				
Without	Reference			
With	1.172	0.420	3.052	0.424
COPD				
Without	Reference			
With	1.653	0.222	9.762	0.598
Dyslipidemia				
Without	Reference			
With	0.965	0.242	3.802	0.755
Cancer				
Without	Reference			
With	0.000	-	-	0.744
Hemostasis				
Without	Reference			
With	1.113	0.520	1.973	0.497
EVL				
Without	Reference			
With	1.253	0.635	1.997	0.386
Clopidogrel				
Without	Reference			
With	0.825	0.562	1.342	0.489
Aspirin				
Without	Reference			
With	0.777	0.357	1.241	0.635
Urbanization level				
1 (Highest)	Reference			
2	1.503	0.552	4.097	0.435
3	0.000	-	-	0.999
4 (Lowest)	0.000	-	-	0.999
Level of care				
Hospital center	2.986	0.411	19.560	0.268
Regional hospital	1.872	0.184	11.435	0.562
Local hospital	Reference			

The abbreviations: hazard ratio (HR); confidence interval (CI); esophagogastroduodenoscopy (EGD); New Taiwan dollar (NTD); diabetes mellitus (DM); hypertension (HTN); chronic obstructive pulmonary disease (COPD); endoscopic variceal ligation (EVL).

**Table 3 jcm-11-03765-t003:** CAD recurrence by using multivariable Cox regression during the tracking period.

Tracking Period	EGD	Adjusted HR	95% CI	95% CI	*p*
Overall (in 1 month)	Without	Reference			
	With	0.855	0.793	1.352	0.411
In 3 weeks	Without	Reference			
	With	0.834	0.693	2.111	0.653
In 2 weeks	Without	Reference			
	With	0.775	0.601	3.075	0.751
In 1 week	Without	Reference			
	With	0.712	0.567	4.235	0.850

The abbreviations: coronary artery disease (CAD); esophagogastroduodenoscopy (EGD); hazard ratio (HR); confidence interval (CI).

**Table 4 jcm-11-03765-t004:** Other covariates of outcomes in 1 month by using multivariable Cox regression.

	EGD	Adjusted HR	95% CI	95% CI	*p*
ICU	Without	Reference			
	With	1.298	0.796	1.896	0.303
Mechanical ventilation	Without	Reference			
	With	1.134	0.675	1.813	0.486
Vasoconstrictors	Without	Reference			
	With	1.560	0.865	2.204	0.762

The abbreviations: esophagogastroduodenoscopy (EGD); hazard ratio (HR); confidence interval (CI); intensive care unit (ICU).

**Table 5 jcm-11-03765-t005:** Days to CAD recurrence in one month.

EGD	Min	Median	Max	Mean ± SD
With	1.00	19.50	21.00	19.50 ± 2.12
Without	1.00	14.00	29.44	14.93 ± 9.96
Total	1.00	15.44	29.44	15.47 ± 9.45

The abbreviations: coronary artery disease (CAD); esophagogastroduodenoscopy (EGD); standard deviation (SD).

## Data Availability

All data generated or analyzed during this study are included in this published article.

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
