# Peer review of "The Clinical Dilemma of Esophagogastroduodenoscopy for Gastrointestinal Bleeding in Cardiovascular Disease Patients: A Nationwide-Based Retrospective Study"

_jcm, 2022, doi:10.3390/jcm11133765_

Round 1
Reviewer 1 Report
In the large retrospective cohort, the authors described peri-procedural complications (e.g., CAD recurrence, death) on performing esophagogastroduodenoscopy (EGD). It is an interesting article; however, there are some concerns in this article. 1. There is no information on anticoagulant use (e.g., DOAC, warfarin, and antiplatelets) 2. There is no information on procedures for therapeutic EGD(e.g., hemostasis) 3. What kinds of upper gastrointestinal scope for available in this cohort? 4. The reason for including or excluding criteria is not apparent. What is the definition of UGIB? Did they exclude small bowel bleeding? 5. Is diabetes including T1DM? The authors should define each variable. 6. There is a randomized controlled trial in a similar cohort. ( e.g., Scientific Reports | (2022) 12:5798 | https://doi.org/10.1038/s41598-022-09911-5) on this issue. 6. References are small and old.
Author Response
June 20, 2022
Co-Editor-in-Chief
Journal of Clinical Medicine
Manuscript entitled " A clinical dilemma of esophagogastroduodenoscopy for gastrointestinal bleeding in cardiovascular disease patients: A nationwide-based retrospective study "
Dear Editor-in-Chief:
Thank you for giving us the opportunity to revise the article " A clinical dilemma of esophagogastroduodenoscopy for gastrointestinal bleeding in cardiovascular disease patients: A nationwide-based retrospective study ". We have made amendments to the manuscript in accordance with the reviewer’s suggestions.
For Reviewer #1
- There is no information on anticoagulant use (e.g., DOAC, warfarin, and antiplatelets)
Re: Thank you for your valuable suggestions. Variables including acute CAD antiplatelet medication: clopidogrel and aspirin are analyzed in table 1. The antiplatelet agents use between the two groups (with and without EGD) showed no significance (p = 0.172 and 0.221). We made amendments in the manuscript. Please see lines 111-113, table 1 and 2.
- There is no information on procedures for therapeutic EGD (e.g., hemostasis)
Re: Thank you for your valuable suggestions. Variables including therapeutic managements: are analyzed in table 1. There was no statistical difference in therapeutic variables: hemostasis and esophageal varices ligation (p = 0.560 and 0.844). We made amendments in the manuscript. Please see lines 111-113, table 1 and 2.
- What kinds of upper gastrointestinal scope for available in this cohort?
Re: Thank you for your valuable suggestions. Because this is a nationwide-based retrospective study, there is no available information about kinds of upper gastrointestinal scope. We will take this issue into consideration in the future study.
- The reason for including or excluding criteria is not apparent. What is the definition of UGIB? Did they exclude small bowel bleeding?
Re: Thank you for your helpful suggestions. The definition of the occurrence of UGIB during hospital stay were obtained from administrative claims (ICD-9-CM 53X.0, 53X.2, 53X.4, 53X.6 (X = 1–4), 535.X1). The diagnosis of small bowel bleeding is not included in our study population. We made amendments in the manuscript. Please see lines 76-82.
- Is diabetes including T1DM? The authors should define each variable.
Re: Thank you for your helpful suggestions. The comorbidities of diabetes mellitus (DM) (ICD-9-CM 250) enrolled in our study means ICD-9-CM Diagnosis Code 250.00
(Diabetes mellitus without mention of complication, type II or unspecified type, not stated as uncontrolled). We did not consider about ICD-9-CM Diagnosis Code 250.01
(Diabetes mellitus without mention of complication, type I [juvenile type], not stated as uncontrolled) With regard to the above description, we made amendments in the manuscript. Please see lines 89, 106, 127, 132, table 1 and 2.
- There is a randomized controlled trial in a similar cohort. ( e.g., Scientific Reports | (2022) 12:5798 | https://doi.org/10.1038/s41598-022-09911-5) on this issue. References are small and old.
Re: Thank you for your valuable suggestions. We reviewed the randomized controlled trial ( e.g., Scientific Reports | (2022) 12:5798 |https://doi.org/10.1038/s41598-022-09911-5) and latest publication associated with this issue. We made amendments in the manuscript and cited new references. Please see references 5, 12, 15 and 16.
Thank you for your valuable suggestions. We feel that the manuscript had improved substantially as a result of the suggestions from your expert reviewers. A revised version of the manuscript has been uploaded together with this cover letter to the Author Center of your Journal.
Sincerely,
Dr. Chao-Feng Chang
Division of Gastroenterology, Department of Internal Medicine, Tri-Service General Hospital, National Defense Medical Center, Taipei, Taiwan, ROC
No. 325, Cheng-Kung Road, Sec. 2, Neihu 114, Taipei, Taiwan
Tel: +886-2-87927208 Fax: +886-2-87927209
E-mail address: taiwanvincent777@gmail.com

Reviewer 2 Report
This is an interesting study using national claim data on whether endoscopy in patients with coronary artery disease is associated with coronary artery disease recurrence or exacerbation.
The biggest problem of this study is that there is no mention of how the primary endpoint, coronary artery disease recurrence, was defined. If the recurrence of coronary artery disease is defined with ICD code, it is difficult to see it as a recurrence of coronary artery disease. This is because the recent coronary artery disease-related ICD code can be repeatedly entered. It is recommended that the combination of additional codes, such as coronary angiography and medications, be defined as coronary artery disease recurrence.
Investigation of other covariates that may associate to recurrence of coronary artery disease is warranted. It can be difficult to use blood or imaging tests, but you need to find a way to represent the severity. For example, whether patients were admitted to an intensive care unit, mechanical ventilation, or use of vasoconstrictors can indirectly indicate severity.
Author Response
June 20, 2022
Co-Editor-in-Chief
Journal of Clinical Medicine
Manuscript entitled " A clinical dilemma of esophagogastroduodenoscopy for gastrointestinal bleeding in cardiovascular disease patients: A nationwide-based retrospective study "
Dear Editor-in-Chief:
Thank you for giving us the opportunity to revise the article " A clinical dilemma of esophagogastroduodenoscopy for gastrointestinal bleeding in cardiovascular disease patients: A nationwide-based retrospective study ". We have made amendments to the manuscript in accordance with the reviewer’s suggestions.
For Reviewer #2
This is an interesting study using national claim data on whether endoscopy in patients with coronary artery disease is associated with coronary artery disease recurrence or exacerbation.
The biggest problem of this study is that there is no mention of how the primary endpoint, coronary artery disease recurrence, was defined. If the recurrence of coronary artery disease is defined with ICD code, it is difficult to see it as a recurrence of coronary artery disease. This is because the recent coronary artery disease-related ICD code can be repeatedly entered. It is recommended that the combination of additional codes, such as coronary angiography and medications, be defined as coronary artery disease recurrence.
Investigation of other covariates that may associate to recurrence of coronary artery disease is warranted. It can be difficult to use blood or imaging tests, but you need to find a way to represent the severity. For example, whether patients were admitted to an intensive care unit, mechanical ventilation, or use of vasoconstrictors can indirectly indicate severity.
Re: Thank you for your valuable recommendation.
- In our study, the recurrence of coronary artery disease is defined with ICD code; however, the recent coronary artery disease-related ICD code might be repeatedly entered. We use therapeutic code such as cardiac catheterization /PTCA in one month to confirm the recurrence of coronary artery disease.
- Other factors: whether patients were admitted to an intensive care unit, mechanical ventilation, or use of vasoconstrictors are further analyzed in our study, and no significant difference was found in initial characteristics (table 1). As to the representation of the severity of our study by using multivariable Cox regression, increasing adjusted HR of ICU, Mechanical Ventilation and Vasoconstrictors use were found in receiving EGD groups (adjusted HR = 1.298, 1.134 and 1.560). However, there is no statistical significance (p = 0.303, 0.486 and 0.762, respectively) (table 4).
We made amendments in the manuscript. Please see lines 97-99, 121-124, 148-153, 280-285, table 1 and 4.
Table 4. Other covariates of outcomes in 1 month by using multivariable Cox regression
|
|
EGD |
Adjusted HR |
95% CI |
95% CI |
P |
|
ICU |
Without |
Reference |
|
|
|
|
With |
1.298 |
0.796 |
1.896 |
0.303 |
|
|
Mechanical Ventilation |
Without |
Reference |
|
|
|
|
With |
1.134 |
0.675 |
1.813 |
0.486 |
|
|
Vasoconstrictors |
Without |
Reference |
|
|
|
|
|
With |
1.560 |
0.865 |
2.204 |
0.762 |
Thank you for your valuable suggestions. We feel that the manuscript had improved substantially as a result of the suggestions from your expert reviewers. A revised version of the manuscript has been uploaded together with this cover letter to the Author Center of your Journal.
Sincerely,
Dr. Chao-Feng Chang
Division of Gastroenterology, Department of Internal Medicine, Tri-Service General Hospital, National Defense Medical Center, Taipei, Taiwan, ROC
No. 325, Cheng-Kung Road, Sec. 2, Neihu 114, Taipei, Taiwan
Tel: +886-2-87927208 Fax: +886-2-87927209
E-mail address: taiwanvincent777@gmail.com
Round 2
Reviewer 1 Report
The article was revised well, and it has improved.
Author Response
June 23, 2022
Co-Editor-in-Chief
Journal of Clinical Medicine
Manuscript entitled " A clinical dilemma of esophagogastroduodenoscopy for gastrointestinal bleeding in cardiovascular disease patients: A nationwide-based retrospective study "
Dear Editor-in-Chief:
Thank you for giving us the opportunity to revise the article " A clinical dilemma of esophagogastroduodenoscopy for gastrointestinal bleeding in cardiovascular disease patients: A nationwide-based retrospective study ". Thank you for your valuable suggestions. We feel that the manuscript had improved substantially as a result of the suggestions from your expert reviewers. A revised version of the manuscript has been uploaded together with this cover letter to the Author Center of your Journal.
Sincerely,
Dr. Chao-Feng Chang
Division of Gastroenterology, Department of Internal Medicine, Tri-Service General Hospital, National Defense Medical Center, Taipei, Taiwan, ROC
No. 325, Cheng-Kung Road, Sec. 2, Neihu 114, Taipei, Taiwan
Tel: +886-2-87927208 Fax: +886-2-87927209
E-mail address: taiwanvincent777@gmail.com

Reviewer 2 Report
After the first revision, the quality of the paper has improved a lot. But I still have some questions.
1. We ask that you add a paragraph on statistical analysis to the Methods section. In particular, please describe your multivariate analysis method in detail in the Methods sections. It is important which covariates are included in multivariable analysis, so please be specific.
2. In line 102. 218 and 872 people had a gastroscopy, aren't 872 patients without an endoscopy?
3. line 83. Please change 16482 to 16,482 as shown in Figure 1. Please correct other numbers as well.
4. Table 1 states that a subset of the patient group who did not undergo endoscopy underwent hemostasis, EVL. is this possible?
Author Response
June 23, 2022
Co-Editor-in-Chief
Journal of Clinical Medicine
Manuscript entitled " A clinical dilemma of esophagogastroduodenoscopy for gastrointestinal bleeding in cardiovascular disease patients: A nationwide-based retrospective study "
Dear Editor-in-Chief:
Thank you for giving us the opportunity to revise the article " A clinical dilemma of esophagogastroduodenoscopy for gastrointestinal bleeding in cardiovascular disease patients: A nationwide-based retrospective study ". We have made amendments to the manuscript in accordance with the reviewer’s suggestions.
For Reviewer #2
After the first revision, the quality of the paper has improved a lot. But I still have some questions.
1.We ask that you add a paragraph on statistical analysis to the Methods section. In particular, please describe your multivariate analysis method in detail in the Methods sections. It is important which covariates are included in multivariable analysis, so please be specific.
Re: Thank you for your valuable recommendation. We made amendments in the manuscript. Please see lines 100-112.
2.3. Statistical analysis
Statistical analysis. We conducted all analyses by SPSS software (version 18; SPSS, Inc., Chicago, IL, USA). The χ2 and Fisher's exact test was used for analysis of categorical variables, such as sex and comorbidities. The Student's t‑test was used for continuous variable, such as age, and the data was presented as mean ± standard error of the mean. Multivariate Cox regression was used to adjust the independent variables and to determine the association between each variable and CAD recurrence in one month. Also, CAD recurrence with different tracking period and other covariates of outcomes were further analyzed by multivariable Cox regression stratified by EGD. Hazard ratio (HR) and 95% confidence interval (CI) were used to evaluate the relative risks between each variable. Mean ± standard error of days to CAD recurrence were further investigated between two groups. The Kaplan‑Meier method was conducted to identify the cumulative survival of CAD with UGI bleeding to determine the statistical significance between groups.
- In line 102. 218 and 872 people had a gastroscopy, aren't 872 patients without an endoscopy?
Re: Thank you for your valuable correction. We corrected the numbers of patients in the paragraph. Please see lines 116-117.
- line 83. Please change 16482 to 16,482 as shown in Figure 1. Please correct other numbers as well.
Re: Thank you for your valuable suggestion. We corrected the numbers in the paragraph. Please see line 83.
- Table 1 states that a subset of the patient group who did not undergo endoscopy underwent hemostasis, EVL. is this possible?
Re: Thank you for your valuable suggestion. In our clinical practice, EGD would be arranged while the patient had the impression of UGIB. There were possible circumstances that clinician might confront. For example:
- Obvious UGIB lesion such as visible vessel with spurting or oozing was observed, and immediate therapeutic management like hemostasis or EVL should be done.
- UGIB with healing ulcer condition, hemorrhagic gastritis, or no active bleeding condition was found under EGD. There was no indication for hemostasis or EVL.
- UGIB with possible malignancy condition and therapeutic management like hemostasis or EVL is ineffective.
Therefore, it was possible that hemostasis or EVL was not indicated for all patients with the diagnosis of UGIB.
Thank you for your valuable suggestions. We feel that the manuscript had improved substantially as a result of the suggestions from your expert reviewers. A revised version of the manuscript has been uploaded together with this cover letter to the Author Center of your Journal.
Sincerely,
Dr. Chao-Feng Chang
Division of Gastroenterology, Department of Internal Medicine, Tri-Service General Hospital, National Defense Medical Center, Taipei, Taiwan, ROC
No. 325, Cheng-Kung Road, Sec. 2, Neihu 114, Taipei, Taiwan
Tel: +886-2-87927208 Fax: +886-2-87927209
E-mail address: taiwanvincent777@gmail.com
